# Clinical and laboratory presentation of first-time antenatal care visits of pregnant women in Ghana, a hospital-based study

Vital Glah Abuku[1]☯, Emmanuel Alote Allotey[1]☯, Maxwell Akonde[2]☯ *

**1** Department of Medical Laboratory Sciences, School of Allied Health Sciences, University of Health and Allied Sciences, Ho, Ghana, **2** Department of Epidemiology and Biostatistics, University of South Carolina Arnold School of Public Health, Columbia, SC, United States of America

☯ These authors contributed equally to this work.
* makonde@email.sc.edu

**Data Availability Statement:** All relevant data are within the paper and its Supporting Information file.

## Abstract

### Background

The WHO recommends pregnant women attend antenatal clinic at least three times during pregnancy; during the first, second and third trimesters. During these visits, an array of clinical and laboratory tests is conducted. The information obtained plays an important role not only in the management and care of pregnancy, but also guides policies targeted at addressing pregnancy-induced health challenges. This study therefore presents laboratory and clinical information of pregnant women at their first antenatal visits.

### Methods

The study was cross-sectional in design which retrospectively reviewed laboratory and clinical data of pregnant women attending their first antenatal clinic (ANC) at the Comboni Hospital, Volta region, Ghana. The data reviewed included information on hemoglobin level, hemoglobin phenotype, malaria diagnostics, Human Immunodeficiency Virus test (HIV), glucose-6-phosphate dehydrogenase (G6PD) deficiency, Hepatitis C Virus (HCV) test, Hepatitis B Virus (HBV) test, Syphilis test, blood pressure, age, urine glucose, and urine protein. The hemoglobin level was assayed with a hemoglobinometer. Qualitative lateral flow chromatographic immunoassay techniques were used to diagnose the HIV, HCV, HBV, syphilis, and malaria status of the pregnant women. Urine dipstick was used assay for the urine protein and urine glucose, whilst the methemoglobin test was used for the G6PD deficiency and alkaline hemoglobin electrophoresis for hemoglobin phenotype. Data on demographic, anthropometric and vital signs such as age, weight and blood pressure were also collected. Descriptive statistics were performed. Frequency and percentages were used to describe the categorical variables and means and standard deviations used to describe the continuous variables.

**Funding:** The authors received no specific funding for this work.

**Competing interests:** The authors have declared that no competing interests exist.

## Results

Hemoglobin S(Hb S) was found in 12.8% of the women with 73.4% having hemoglobin levels below 11.5g/dl. On G6PD deficiency, 1.6% and 0.8% were partially and fully defective respectively. Also, urine protein (1.2%) and glucose (0.4%) were detected. The prevalence of HBV, HCV and malaria were 4.4%, 3.6% and 2.4%, respectively.

## Conclusion

Anemia in pregnancy was high among the study sample. Malaria and hepatitis infections were observed in the study sample. Policies on maternal health should be targeted at providing better nutritional options, that can enhance the hemoglobin level during pregnancy. Pregnant women should benefit from enhanced surveillance for HIV, HBV, HCV, and syphilis.

## Introduction

Pregnancy is characterized by many physiological changes, which may be pathological in the non-pregnant states [1, 2]. Physiologically, the body undergoes a series of adaptive changes to prepare for fetal hematopoiesis and also cushion the body against expected blood loss at delivery [3, 4]. Early commencement of antenatal care by pregnant women as well as regular visits has the potential to affect maternal and fetal outcome positively. Good maternal care is vital for the health of the mother and the development of the unborn baby [5]. The clinical characteristics and laboratory results of pregnant women are used to monitor and to provide good care to the woman and fetus throughout the pregnancy and at delivery and post-delivery. Inadequate care during this time breaks a critical link in the continuum of care and affects both women and babies [5–7].

Traditionally, antenatal care (ANC) programs have been recommended for developing countries along the lines of those used in developed countries, with only minor adjustments for local conditions [5, 8]. Many of the components of these antenatal programs have not been subjected to rigorous scientific evaluation to determine their effectiveness. Despite a widespread desire to improve maternal care services, the inadequacy of evidence has impeded the identification of effective interventions and thus the optimal allocation of resources [6]. The clinical and laboratory data is needed to better direct resources to where improvement is needed. This type of data on pregnant women can help drive policy changes including reallocation of resources towards addressing any maternal and child health gaps. This study aimed to describe the clinical and laboratory presentation of pregnant women during their first-time antenatal clinic visit, and in so doing identify health challenges as they relate to maternal health in Ghana.

## Methods

Clinical and laboratory data of pregnant women who visited the Comboni Hospital located in the Volta region of Ghana, between June 2018 and September 2018 were retrospectively collected for this cross-sectional study. Comboni hospital is a 50-bed facility located in the South Tongu district of the Volta region of Ghana, a region with 1.6million people (According to the 2021 Ghana Population and Housing Census). The hospital provides services to clients from the Southern part of the region and attends to referral cases from nearby districts such as

North Tongu, Ketu North and Keta Municipality. The facility attends to antenatal cases two days (Wednesday and Friday) in a week and attends to averagely 20 first-timers per week. Access to the completely de-identified data through the hospital's electronic system was granted for the purpose of this study in November 2018. The study was limited to pregnant women who visited the hospital for their first-time antenatal clinic services and who were 18years and above. Data on hemoglobin level, hemoglobin phenotype, malaria, HIV, G6PD deficiency, HCV, HBV, syphilis, blood pressure, urine glucose and protein, age and weight were collected. Hemoglobin level, blood pressure, age and weight were recorded as continuous variables. The hemoglobin level was assayed with a urit12® hemoglobinometer (URIT Medical Electronics Co., Ltd. Guangxi, China). Using the hemoglobin level, anemia in this study was defined as hemoglobin level below 11.5g/dl[9] and hence, hemoglobin level was dichotomized into less than 11.5g/dl and 11.5g/dl or more. It was further recorded into severe anemia (Hb<7.0g/dl), moderate anemia (Hb = 7.0–9.9g/dl), mild anemia (Hb = 10.0–11.4g/dl) and normal hemoglobin (Hb> = 11.5g/dl)[9]. HBV, HCV, HIV, and syphilis were performed with the Onsite® rapid test kit, which applies a lateral flow chromatographic immunoassay for qualitative detection of hepatitis B surface antigen and antibodies to HIV-1, HIV-2, HCV and *Treponema pallidum* in human serum or whole blood. Malaria testing was also performed using the same lateral flow technique, but with the CareStart® test kit which detected the Malaria histidine-rich protein 2 (HRP2) or parasite lactate dehydrogenase (pLDH). A confirmation test for malaria parasites was done with blood film microscopy for samples that tested positive with the lateral flow technique. The HBV, HCV, HIV, syphilis, and malaria status were reported as positive for pregnant women who had the infection and negative for those who did not. Urine dipstick was used to assay for the urine protein and urine glucose. Urine glucose and protein were reported as positive, trace and negative. The sodium metabisulphite test was used to test for the sickling status of each participant before phenotyping. Alkaline hemoglobin electrophoresis was performed to determine the hemoglobin phenotype using the hemoglobin A, F, S and C for the control bands. The methemoglobin reduction method, where through the action of nitrite, hemoglobin is oxidized to methemoglobin and in the presence of methylene blue, methemoglobin is reduced back to hemoglobin through the oxidative pathway, was used. G6PD is the enzyme needed for the reduction of methemoglobin and in its absence the solution remains brown. The test was performed by putting 2ml of participants blood in three tubes labeled "positive", "negative", and "test". In the tubes labeled test and positive, 0.1ml of sodium nitrite-glucose solution was added and methylene blue additionally added to the tube labeled test and the solutions mixed gently. No reagent was added to the tube labeled negative. All the tubes were incubated for 3 hours, and then 0.1ml of the solutions of each of the tubes were taken into newly labeled tubes and the volumes were made up to 10ml using normal saline. The color and clear nature of the solution in the tubes labeled test were compared to the positive and negative. The test tube solution that were clear red and similar to the tubes labeled negative are reported as no defect, those that were brown and similar to the tubes labeled positive were reported as full defect and those intermediates were reported partial defect. The study was approved by the University of Health and Allied Sciences (UHAS) Research Ethics Committee (UHAS-REC). Participants' informed consent was waived as only de-identified data was used. Additionally, the Comboni hospital ethics committee granted an approval to access the participants' data. We conducted a complete case analysis. Summary statistics were used to describe the demographic and clinical characteristics of the participants. Categorical variables were described with frequencies and percentages whilst continuous variables were described with means and standard deviations. The positivity rate of infections was compared between women who had hemoglobin level below 11.5g/dl and women who had hemoglobin level of 11.5g/dl or more using chi-square statistics. The

demographic and clinical characteristics were also described across different categories of hemoglobin level (severe anemia (Hb<7.0g/dl), moderate anemia (Hb = 7.0–9.9g/dl), mild anemia (Hb = 10.0–11.4g/dl) and normal hemoglobin (Hb> = 11.5g/dl)). A p-value less than 0.05 was considered statistically significant. The data was analyzed using the Statistical Package for Social Sciences (IBM SPSS), version 22.0.

## Results

Table 1 presents the demographic and clinical characteristics of the women. A total of 250 women with a mean age of 27.8±6.5years participated in the study. The average weight of the participants was 67.2±14.3 kg with mean systolic and diastolic blood pressures (BP) of 107.9 ±10.8 mmHg and 65.2±9.6 mmHg, respectively.

Among the pregnant women, 12.8% had tested positive for sickling. Most of those who tested positive for sickling (10.4%) had hemoglobin AS (HbAS) with 0.4% having hemoglobin SC (HbSC). About 2.4% of the women had some form of G6PD deficiency (0.8% were full defective and 1.6% were partial defective). Urine protein and glucose were detected in a few of the pregnant women, with 4.4% testing positive for hepatitis B virus. HIV and malaria in pregnancy was found in 2.0% and 2.4% of the women, respectively (Table 1).

Anemia was defined by hemoglobin level below 11.5g/dl. 73.6% of the women were anemic (Fig 1). Infections were more common in the women who were anemic compared to those who had normal hemoglobin level (Malaria: 83.3% vs 16.7%; HIV: 80.0% vs 20.0%; syphilis: 62.5% vs 37.5%; HCV: 66.7% vs 33.3%). All the women who had HBV were anemic (Table 2).

Most of the women (85.3%) who were anemic had hemoglobin AA (HbAA) followed by hemoglobin AS (HbAS) (12.0%). Among those with normal hemoglobin levels, 92.4% had HbAA, with the rest being either HbAS (6.1%) or HbAC (1.5%) (Fig 2).

The women were again categorized as Severe anemia (Hb<7.0g/dl), Moderate anemia (Hb = 7.0–9.9g/dl), Mild anemia (Hb = 10.0–11.4g/dl) and Normal hemoglobin (Hb> = 11.5g/dl). On average, the pregnant women with severe anemia were the youngest whilst those with normal hemoglobin were the oldest. Most of the women who tested positive for sickling had moderate anemia (Table 3). The average weight followed a similar trend with the severe anemic group having lower weights and normal hemoglobin group being the heavier. All the groups had normal blood pressure for the systolic and diastolic.

## Discussion

Clinical and laboratory data of pregnant women are important for the management and prevention of pregnancy-associated complications as well as driving maternal and child health policies. This study presented the clinical and laboratory data of pregnant women attending their first antenatal clinic in a hospital in Ghana. Overall, there has been improvement in certain areas of maternal health in the last two decades such as malaria in pregnancy, whilst other areas such as anemia in pregnancy needs serious attention [9, 10].

Anemia was predominant in the study population with over 73% having their hemoglobin below 11.5g/dl. This is more than the average of 56% prevalence of anemia in developing countries [10, 11]. Pregnancy is a stressful physiological condition that has negative effects on hemoglobin production [3, 4]. Nutrients such as folic acid and iron that are needed for hemoglobin production, are equally utilized by the fetus for development [12–14]. This explains why the expected (average) hemoglobin level during pregnancy is lower compared to nonpregnant women. However, anemia during pregnancy has serious consequences on both the fetus and the pregnant woman. Anemia in pregnancy is associated with adverse pregnancy outcomes including miscarriage and preterm birth [15–17]. The World Health organization

**Table 1. Summary of demographic and clinical characteristics of the pregnant women.**

| Characteristic | Total(N = 250) |
|---|---|
| Weight(kg) | 67.2±14.3 |
| **Age(years)** | 27.8±6.5 |
| **Blood Pressure** | |
| **Systolic (mmHg)** | 107.9±10.8 |
| **Diastolic (mmHg)** | 65.2±9.6 |
| **Hemoglobin (g/dl)** | 10.3±1.63 |
| Sickling Status | |
| Positive | 32(12.8) |
| Negative | 218(87.2) |
| Hemoglobin variants | |
| AA* | 218(87.2) |
| AS | 26(10.4) |
| SC | 1(0.4) |
| AC | 5(2.0) |
| G6PD | |
| Normal | 244(97.6) |
| Partial | 4(1.6) |
| Full defect | 2(0.8) |
| Urine Protein | |
| Negative | 226(90.4) |
| Trace | 21(8.4) |
| Positive | 3(1.2) |
| Urine Glucose | |
| Negative | 247(98.8) |
| Trace | 2(0.8) |
| Positive | 1(0.4) |
| Malaria | |
| Positive | 6(2.4) |
| Negative | 244(97.6) |
| Syphilis | |
| Positive | 8(3.2) |
| Negative | 242(96.8) |
| HIV | |
| Positive | 5(2.0) |
| Negative | 245(98.0%) |
| HCV | |
| Positive | 9(3.6) |
| Negative | 241(96.4) |
| HBV | |
| Positive | 11(4.4) |
| Negative | 239(95.6) |

Data is presented as frequencies with percentages of the respective parameters in parenthesis for the categorical variables and mean ± standard deviation for the continuous variables. * Only one hemoglobin band was seen but read as two.

recommends regular folic acid and vitamin B12 supplement during pregnancy [15, 18, 19]. In Ghana, this recommendation has been implemented by the Ghana health service since 2002,

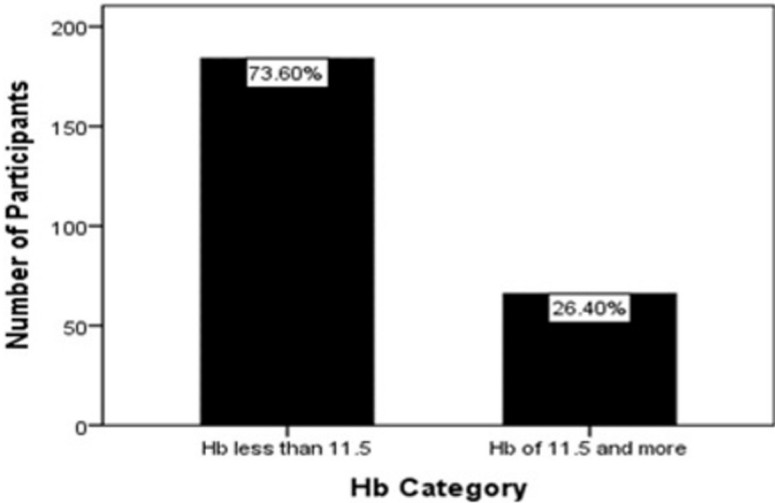

**Fig 1. Incidence of anemia among the pregnant women.**

and most pregnant women who attend antenatal care are provided with the prescriptions [20, 21]. Even though, these were covered under the national health insurance scheme and the free maternal and child health policies of the government of Ghana, equitable resource allocation has hampered its effective implementation. Hence, pregnant women are directed to buy the supplements from private pharmacies [20, 22]. Given that the study site is located in a low-income community, one explanation for the high proportion of anemia among the women is the low socioeconomic status. This also has consequences on the quality of food the women eat during pregnancy. Additionally, the nutritional status of the pregnant women may have been impacted by their diet before the pregnancy. However, this study could not assess the dietary information before pregnancy. Our study results were comparable to the findings by Acheampong et al., [23] and Pobee et al., [24] who found the prevalence of anemia in pregnancy to be more than 50% and 63% in Ghana, respectively and Awoyesku et al., [25] who found 60% of anemia in pregnant women in Port Harcourt, Nigeria. The results were however contrary to some other studies conducted in Switzerland (18.5%) [26] and China (19.9%) [27], emphasizing the relationship between income level and anemia in pregnancy.

The proportion of pregnant women who tested positive for malaria was 2.4%. Malaria in pregnancy has been on the decline in Ghana [28]. This is primarily due to the campaign at preventing malaria in pregnancy. Programs such as indoor residual spraying, the distribution and

**Table 2. Distribution of infection positivity among the women stratified by hemoglobin level (Hb<11.5 and Hb≥11.5g/d).**

|  | Hb <11.5g/dl | Hb≥11.5g/dl | p-value |
|---|---|---|---|
| Malaria | 5(83.3) | 1(16.7) | 0.584 |
| HIV | 4(80.0) | 1(20.0) | 0.743 |
| Syphilis | 5(62.5) | 3(37.5) | 0.469 |
| HBV | 11(100.0) | 0(0.0) |  |
| HCV | 6(66.7) | 3(33.3) | 0.631 |

Data is presented as frequencies with percentages of the respective parameters in parenthesis and the significant level at 0.05.

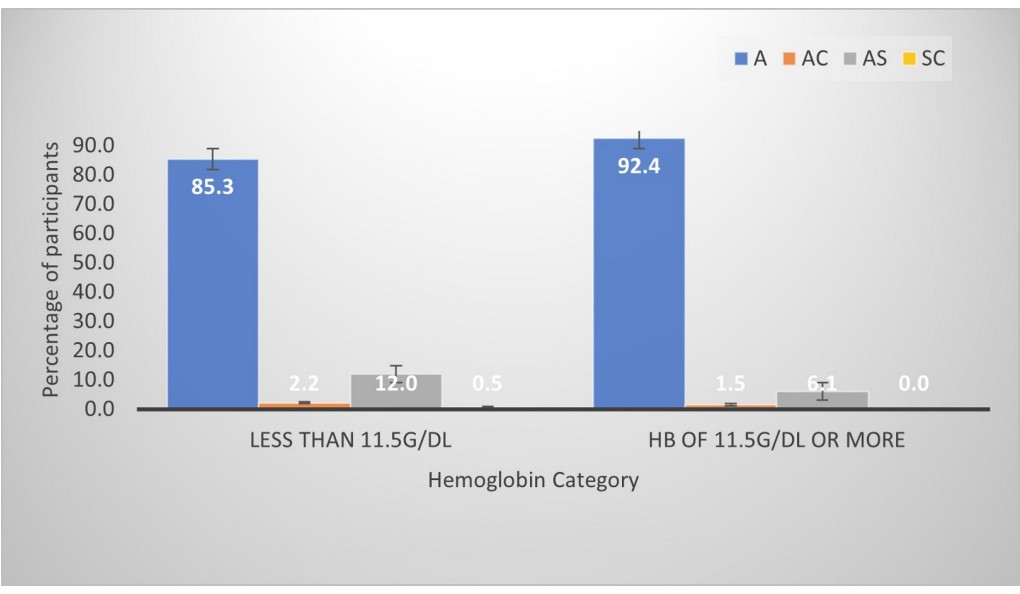

**Fig 2. Incidence of anemia among the various hemoglobin variants observed in the subjects.**

use of insecticide treated bed nets, and other preventive measures against malaria have been directed at preventing malaria in pregnancy [29, 30]. The study results reflect a reduction in malaria in pregnancy when compared with previous studies [28], even though more need to

**Table 3. Demographic characteristics and laboratory presentation of the pregnant women stratified into severe anemia (Hb<7.0g/dl), moderate anemia (Hb = 7.0–9.9g.dl), mild anemia (Hb = 10.0–11.4g/dl) and normal hemoglobin (Hb> = 11.5g/dl).**

| | TOTAL | SEVERE ANEMIA | MODERATE ANEMIA | MILD ANEMIA | NORMAL HEMOGLOBIN |
|---|---|---|---|---|---|
| **Total** | 250(100) | 8(3.2) | 95(38.0) | 81(32.4) | 66(26.4) |
| **Age** | | 23.8±7.8 | 26±6.6 | 28.3±6.6 | 29.9±5.2 |
| **Blood pressure** | | | | | |
| Systolic | | 102.5±10.4 | 105.3±11.0 | 107.9±10.3 | 112.3±10.0 |
| Diastolic | | 60.0±9.3 | 63.0±8.4 | 65.2±8.7 | 69.2±10.9 |
| **Weight** | | 61.1±12.9 | 64.9±12.7 | 65.7±13.7 | 73.0±15.6 |
| **Hemoglobin level** | | 6.5±0.5 | 8.9±0.8 | 10.7±0.5 | 12.2±0.6 |
| **Sickling** | | | | | |
| Negative | 218(87.2) | 7(87.5) | 80(84.2) | 70(86.4) | 61(92.4) |
| Positive | 32(12.8) | 1(12.5) | 15(15.8) | 11(32.4) | 5(7.6) |
| **G6PD Status** | | | | | |
| Normal | 244(97.6) | 8(100.0) | 93(97.9) | 78(96.3) | 65(98.5) |
| Full Defect | 2(0.8) | 0(0.0) | 1(1.1) | 1(1.2) | 0(0.0) |
| Partial Defect | 4(1.6) | 0(0.0) | 1(1.1) | 2(2.5) | 1(1.5) |
| **Hemoglobin Variant** | | | | | |
| A | 218(87.2) | 7(87.5) | 80(84.2) | 70(86.4) | 61(92.4) |
| AS | 26(10.4) | 1(12.5) | 12(12.6) | 9(11.1) | 4(6.1) |
| SC | 1(0.4) | 0(0.0) | 1(1.1) | 0(0.0) | 0(0.0) |
| AC | 5(2.0) | 0(0) | 2(2.1) | 2(2.5) | 1(1.5) |

Data is presented in the form of mean ± standard deviation for continuous variables and frequencies with percentages of the respective parameters in parenthesis for categorical variables; A, S, and C represented hemoglobin A, S, and C, respectively.

be done to protect the pregnant women from malaria. There is the need for a further study on the adherence level to malaria intervention programs between the women who had malaria and those who did not. Malaria parasites cause the lysis of red blood cells and are major risk factor of anemia in pregnancy, and hence adverse pregnancy outcomes [31, 32]. This is reflected in our study as 83.3% of the malaria positive cases were the anemic women.

Access to screening for infections such as HIV, syphilis, hepatitis B virus and hepatitis C virus offers pregnant women the opportunity to seek preventive measures including vaccination and provides information to clinicians to institute measures at reducing mother-to-child transmission of these infections. In the current study, the proportion of pregnant women with HIV, syphilis, HBV, and HCV were lower, compared to the prevalence of these infections in the general adult population where the prevalence of HIV, HCV, HBV, and syphilis were 4.9%, 4.4%, 7.5% and 8.4% respectively [33–35]. This may be another success story of the maternal and child health program in Ghana [36–38]. However, among the women with anemia, the prevalence of these infections was high. Compared to women with normal hemoglobin, 80%, 62.5% and 66.7% of the HIV, syphilis, and HCV infections were in the women with anemia. One explanation is that anemia impacts the immune system response and may lead to infections being able to establish themselves easily. The reverse may also be true, where the infectious agents are causing the fast degradation of the red blood cells leading to low hemoglobin in those with the infections. For example, malaria parasites cause the lysis of red blood cells, which can progress to anemia.

## Limitations

The study had some limitations that may affect the interpretations of the results. First, this was a cross-sectional study and hence, limitations associated with cross-sectional studies are applied to this study. Second, the study lacks temporal sequence because this was cross-sectional in design, and we are unable to tell if the existing intervention programs such as malaria control programs led to the clinical and laboratory outlook in these women. Again, the seasonality of malaria may have influenced the findings. Since the study was conducted between June and September when the highest number of malaria cases are reported, the incidence of malaria if the data was collected all year will likely to be lower. Third, we are unable to generalize these results to pregnant women in Ghanaian population, as the women were recruited from a hospital in a particular locality which has no central reach to most pregnant women in Ghana. Additionally, future studies should look at increasing the sample size as 250 is small for a population level inference. Fourth, data on gestation age or trimester in which women attended their first antenatal visit was not collected. This is important as some clinical and laboratory presentations such as hemoglobin level, are affected by the gestational age.

## Conclusion

Despite the limitations, this study provides important findings that require the attention of policy makers. The study indicates anemia remains a challenge among pregnant women despite efforts at making antenatal clinic free and providing supplements during these clinic visits. Additionally, infection rates among pregnant women with anemia are high. Infections, particularly malaria, may lead to loss of red blood cells and hence anemia. Anemia in pregnancy may also be a potential risk factor for infections such as HIV, syphilis, and HCV. Policies on maternal health should be targeted at providing better nutritional options, that can enhance the hemoglobin level during pregnancy. Pregnant women should benefit from enhanced surveillance for HIV, HBV, HCV, and syphilis.

## Supporting information

**S1 Dataset.**
(XLSX)

## Author Contributions

**Conceptualization:** Vital Glah Abuku, Emmanuel Alote Allotey, Maxwell Akonde.

**Data curation:** Vital Glah Abuku.

**Formal analysis:** Emmanuel Alote Allotey, Maxwell Akonde.

**Investigation:** Vital Glah Abuku.

**Methodology:** Emmanuel Alote Allotey.

**Project administration:** Vital Glah Abuku.

**Supervision:** Emmanuel Alote Allotey.

**Writing – original draft:** Vital Glah Abuku, Maxwell Akonde.

**Writing – review & editing:** Emmanuel Alote Allotey, Maxwell Akonde.

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
