## [Decision Letter · Decision Letter 0]

4 May 2022

PONE-D-21-22935Clinical and Laboratory Presentation of First time Antenatal Care Visits of Pregnant Women in Ghana, a Hospital-Based StudyPLOS ONE

Dear Dr. Akonde,

Thank you for submitting your manuscript to PLOS ONE. After careful consideration, we feel that it has merit but does not fully meet PLOS ONE’s publication criteria as it currently stands. Therefore, we invite you to submit a revised version of the manuscript that addresses the points raised during the review process.

In particular, please ensure you address the comment regarding clearly stating the research question and objective of this study in the Introduction section of your manuscript. Please also ensure that you fully address all comments regarding information about patient records and provision of patient consent.

We look forward to receiving your revised manuscript.

Kind regards,

Hugh Cowley

Senior Editor

PLOS ONE

2. In the ethics statement in the manuscript and in the online submission form, please provide additional information about the patient records used in your retrospective study, including: a) whether all data were fully anonymized before you accessed them; b) the date range (month and year) during which patients' medical records were accessed; c) the date range (month and year) during which patients whose medical records were selected for this study sought treatment. If the ethics committee waived the need for informed consent, or patients provided informed written consent to have data from their medical records used in research, please include this information.

Reviewers' comments:

Reviewer's Responses to Questions

**Comments to the Author**

1. Is the manuscript technically sound, and do the data support the conclusions?

Reviewer #1: Partly

2. Has the statistical analysis been performed appropriately and rigorously? 

Reviewer #1: No

3. Have the authors made all data underlying the findings in their manuscript fully available?

Reviewer #1: No

4. Is the manuscript presented in an intelligible fashion and written in standard English?

Reviewer #1: Yes

5. Review Comments to the Author

Reviewer #1: General comments: The manuscript treats an important question about maternal health in a low-income community.

As the data were collected retrospectively in 2018, It could have been great to add information on the outcomes (mother/offspring) associated with the conditions studied.

Abstract

In general, the abstract will need to be modified after the modification of the main text.

Line 35: it will be better to have hemoglobin phenotyping after hemoglobin level on line 33

Line 41- 43: conclusion needs to be improved. The fact that pregnant women were asymptomatic or symptomatic was not stated in the text.

Conclusion highlights only malaria and anemia.

Introduction

The introduction needs to address clearly the research question and the objective of the study

Lines 53-55: please reformulate

Methodology

There is only little information provided in this section.

Study area: please provide general information on the background, rural/urban health services available, uptake of antenatal interventions

The description of the health facility where the data come from is lacking (how many pregnant women attend antenatal care, how many deliveries per month/year?

On average, when do pregnant women start the antenatal care in the region/district? What is/are the main source(s) of income

Nutritional status of the population (preschool children, pregnant women)

Please provide information on altitude, which can affect the interpretation of hemoglobin.

Line72: is the “weight” demographic parameter or anthropometric/clinical?

The authors need to provide information on the assays /platforms used to diagnose the conditions studied HIV, HBV… and the interpretation of the results

Please provide information on the management of those who were positive for HIV, HVB…

I guess the variable “Height” may have been recorded, why it is not part of the data?

Gestational age is important and should be integrated

More information about the cut-off used for anemia is needed (reference…)

I suggest presenting a flow chart showing the process of data collection

Results

General: please use one decimal (line 77, 78,79…)

Information in tables are redundant with those in the comment accompanying tables

All transformation of variables should be indicated in the methodology section (example age)

Table 1: the title of the table stipulate “Demographic characteristic…” but in the text there are clinical and biological data

It is not important to present the mean and the mode, I suggest removing the mode

Table 2. I suggest changing the title. According to the title of the manuscript table 2 should be looking like “Laboratory characteristics of pregnant women …” (just an example)

Information in this table needs to be describe in the methodology (meaning, how the diagnosis was made…)

Line 93: not clear

Line 95: provide the source of the cut-off and put it the methodology

Lines 96-98:

Table 3: the cut-offs of anemia and not-anemia overlap need to be clarified (<=11.5 and >=11.5)

I suggest adding two rows: one with all infections combined and the second presenting subject without infection

It doesn’t seem to be important to have P value for HBV

Give information on the statistical test(s) used

Table 4: the title should be changed, it says demographic characteristics and infections… but there is no infection in the data presented

Table 4 needs legend with explanation for abbreviations

Discussion

Line 120: …important for the management and prevention (please add prevention)

Lines 128: iron needed for red cells production/ hemoglobin?

Lines 133-134: please confirm micronutrients recommended during pregnancy by WHO

Lines 141-142: nutritional status of pregnant women is also affected by the diet before pregnancy

Lines 142-143: Please reformulate the sentence

Lines 144-145: need also to compare with a country (countries) in the west African region

Lines 150: it is not clear what means “…other malaria preventive techniques…”

As malaria diagnosis was done per routine protocol, it is likely that microscopy and/or rapid diagnosis test was/were used; sub microscopy (common during pregnancy) needs to be discussed

Lines 160-161: what is the picture in the general population?

Lines 161-162: what was done to explain the “success story”, what was the previous picture and what is the current in the general population? How representative are these results from one hospital (June-September 2018)?

Lines 162-163: what could be the explanations.

Lines 164-165: the fact that most of the infections studied are blood-borne doesn’t fit as a possible explanation for infected pregnant women to be more anemic. Please revise the pathogenesis of anemia related to malaria and anemia related to infection other than malaria and also anemia associated with chronic infection/inflammation

Lines 166-167: it is not clear whether after transfusion women were still anemic

Line 168: test hypothesis instead of establish hypothesis

Line 175: the authors need to add gestational age on their data; I assume this information is available in the record from the hospital

Lines 181-183: Which could be the risk factor? Is anemia the risk factor for infections or infections are the risk factors for anemia?

6. PLOS authors have the option to publish the peer review history of their article (what does this mean?). If published, this will include your full peer review and any attached files.

Reviewer #1: No

---

## [Author Response · Author response to Decision Letter 0]

2 Jun 2022

I have attached my response to the comments from the reviewer and editor in the file titled "Response to Reviewers".

---

## [Decision Letter · Decision Letter 1]

4 Nov 2022

PONE-D-21-22935R1Clinical and laboratory presentation of first-time antenatal care visits of pregnant women in Ghana, a hospital-based studyPLOS ONE

Dear Dr. Akonde,

Thank you for submitting your manuscript to PLOS ONE. After careful consideration, we feel that it has merit but does not fully meet PLOS ONE’s publication criteria as it currently stands. Therefore, we invite you to submit a revised version of the manuscript that addresses the points raised during the review process.

We look forward to receiving your revised manuscript.

Kind regards,

Orvalho Augusto, MD, MPH

Academic Editor

PLOS ONE

Reviewers' comments:

Reviewer's Responses to Questions

**Comments to the Author**

1. If the authors have adequately addressed your comments raised in a previous round of review and you feel that this manuscript is now acceptable for publication, you may indicate that here to bypass the “Comments to the Author” section, enter your conflict of interest statement in the “Confidential to Editor” section, and submit your "Accept" recommendation.

Reviewer #2: (No Response)

2. Is the manuscript technically sound, and do the data support the conclusions?

Reviewer #2: Partly

3. Has the statistical analysis been performed appropriately and rigorously? 

Reviewer #2: No

4. Have the authors made all data underlying the findings in their manuscript fully available?

Reviewer #2: Yes

5. Is the manuscript presented in an intelligible fashion and written in standard English?

Reviewer #2: Yes

6. Review Comments to the Author

Reviewer #2: The manuscript by Abuku and collaborators aims to “to describe the clinical and laboratory presentation of pregnant women during their first-time antenatal clinic visit, and in so doing identify health challenges as they relate to maternal health in Ghana” is an important issue in health and Global Health, however involving a very small number of pregnant women during a very small period of time (just 4 months). It is a descriptive manuscript lacking some important information concerning the participants and concerning data analysis.

I would emphasize that the reduced study time period unable the authors to describe the seasonality variability in some disease, especially malaria, and that should be mentioned in the limitations of the study.

In line 92, there should be an information concerning the hemoglobinometer used in the study.

Line 94, it misses “into less than 11.5g/dl and (more than) 11.5g/dl.”

Line 110 and 111. The G&PD activity method should be explained, and the reference values for G6PD deficiency should be mentioned.

Line 131. Authors have include HBB C allele as sickling allele, but it is not classified as a sickling allele, it indeed protects from sickling in SC patients. In that sense I would suggest to refer just 10.4 % sickling. The authors does not refer which method they use to test sickling, or if they are using electrophoresis result to consider sickling.

Table 1 -I would suggest to include max and min values in all variables.

There is no information concerning the existence (or not) of multiple infection (or mixed infections) in any pregnant women.

There is no analysis of G6PD def or HBB genotype as a risk factor for infection or other laboratory parameter alteration.

Line 152 and 153 – The comparation is not informative. Authors should determine the frequency in each classes (22/26 and 157/162) and compare those results. These sentence should be rewrite.

There are any result or data in pregnancy complications? An its association with G6PD, HBB, Infections etc.

Discussion

Line 175. In the sentence “Overall, there have been improvement in certain” there is no information in which parameters and compared to what values or periods or regions.

Line 179 – “This is more than the average of 56% prevalence of anemia in developing countries” this is to the general population? And for pregnant women?

Line 189. There is lacking information if the pregnant women from the study use Vitamin supplements.

Lijne 209 – 211 . Please clarify the sentence. Does the authors refers to preventive therapeutics?

Is missing discussion concerning Mixed infections, influence of S allele and C allele, G6PD deficiency.

Limitations – Please clarify sentence in line 231-232

Include as limitation : small sample size, just 4 months of study, doesn’t covering malaria seasonality or food availability

Fig 2 -it should be represented in % by hemoglobin category and not absolute numbers

7. PLOS authors have the option to publish the peer review history of their article (what does this mean?). If published, this will include your full peer review and any attached files.

Reviewer #2: **Yes: **Miguel Brito

---

## [Author Response · Author response to Decision Letter 1]

8 Nov 2022

We have revised the manuscript and responded to the reviewer(s) and editor's comments in the Response to Reviewers document.

---

## [Decision Letter · Decision Letter 2]

6 Dec 2022

PONE-D-21-22935R2Clinical and laboratory presentation of first-time antenatal care visits of pregnant women in Ghana, a hospital-based studyPLOS ONE

Dear Dr. Akonde,

Thank you for submitting your manuscript to PLOS ONE. After careful consideration, we feel that it has merit but does not fully meet PLOS ONE’s publication criteria as it currently stands. Therefore, we invite you to submit a revised version of the manuscript that addresses the points raised during the review process.

We look forward to receiving your revised manuscript.

Kind regards,

Orvalho Augusto, MD, MPH

Academic Editor

PLOS ONE

Journal Requirements:

Reviewers' comments:

Reviewer's Responses to Questions

**Comments to the Author**

1. If the authors have adequately addressed your comments raised in a previous round of review and you feel that this manuscript is now acceptable for publication, you may indicate that here to bypass the “Comments to the Author” section, enter your conflict of interest statement in the “Confidential to Editor” section, and submit your "Accept" recommendation.

Reviewer #3: All comments have been addressed

2. Is the manuscript technically sound, and do the data support the conclusions?

Reviewer #3: Yes

3. Has the statistical analysis been performed appropriately and rigorously? 

Reviewer #3: Yes

4. Have the authors made all data underlying the findings in their manuscript fully available?

Reviewer #3: Yes

5. Is the manuscript presented in an intelligible fashion and written in standard English?

Reviewer #3: Yes

6. Review Comments to the Author

Reviewer #3: 1. Line 40: The method for determination of G6PD status should be methaemoglobin reduction test, and not cyanmethaeglobin (which is used for hemoglobin determination).

2. Methods: Please describe how clinical and laboratory data was collected. Was the data collected from electronic records or from paper folders.

3. Line 90: Continuous should be "continuous variables"

4. Line 99: Please change "malaria was performed" to "malaria testing was performed"

5. Line 101: It is not clear that malaria microscopy was used to confirm RDT results. Is this the normal practice? How is this done (are both tests done simultaneously or it is only RDT positive samples that are further tested with microscopy)?

6. Line 140: Please add the sample size to the percentages, e.g. 12.8% (n/N)

7. Table 1: Please change "hemoglobin electrophoresis" to "hemoglobin variants"

8: Line 208: The anemia prevalence could also be compared to other African countries

9. Please check that all references are complete, e.g. Ref 4,

7. PLOS authors have the option to publish the peer review history of their article (what does this mean?). If published, this will include your full peer review and any attached files.

Reviewer #3: No

---

## [Author Response · Author response to Decision Letter 2]

6 Dec 2022

We have attached a response to the reviewers, but also find below the response:

Reviewer #3: 1. Line 40: The method for determination of G6PD status should be methaemoglobin reduction test, and not cyanmethaeglobin (which is used for hemoglobin determination): We have corrected this and elaborated on the methaemoglobin method used.

2. Methods: Please describe how clinical and laboratory data was collected. Was the data collected from electronic records or from paper folders.: This hospital had an electronic record system. We clarified that by inserting: “Access to the completely de-identified data through the hospitals electronic system was granted for the purpose of this study in November 2018.”

3. Line 90: Continuous should be "continuous variables": “Variables” is inserted

4. Line 99: Please change "malaria was performed" to "malaria testing was performed": “Testing” is inserted.

5. Line 101: It is not clear that malaria microscopy was used to confirm RDT results. Is this the normal practice? How is this done (are both tests done simultaneously or it is only RDT positive samples that are further tested with microscopy)?: Malaria microscopy is done for only RDT positive samples. We added the following sentence to clarify this: “A confirmation test for malaria parasites was done with blood film microscopy for samples that tested positive with the lateral flow technique.” 

6. Line 140: Please add the sample size to the percentages, e.g. 12.8% (n/N): We strongly believe that reporting percentages and the actual counts will amount to repeating the results. We have the actual counts in the tables which makes referencing it when needed easier, however the percentages better describe the results and hence the consistency throughout the results section as seen in manuscripts published by Plos One Journal. 

7. Table 1: Please change "hemoglobin electrophoresis" to "hemoglobin variants": We have changed the “electrophoresis” to “variants”.

8: Line 208: The anemia prevalence could also be compared to other African countries: We added comparative findings from Nigeria to the discussion

9. Please check that all references are complete, e.g. Ref 4,: We checked all references and ensured they were complete. We revised Reference 4 to make it complete.

---

## [Decision Letter · Decision Letter 3]

21 Dec 2022

Clinical and laboratory presentation of first-time antenatal care visits of pregnant women in Ghana, a hospital-based study

PONE-D-21-22935R3

Dear Dr. Akonde,

We’re pleased to inform you that your manuscript has been judged scientifically suitable for publication and will be formally accepted for publication once it meets all outstanding technical requirements.

Kind regards,

Orvalho Augusto, MD, MPH

Academic Editor

PLOS ONE

Additional Editor Comments (optional):

Please do address the comments from the reviewer below.

Reviewers' comments:

Reviewer's Responses to Questions

**Comments to the Author**

1. If the authors have adequately addressed your comments raised in a previous round of review and you feel that this manuscript is now acceptable for publication, you may indicate that here to bypass the “Comments to the Author” section, enter your conflict of interest statement in the “Confidential to Editor” section, and submit your "Accept" recommendation.

Reviewer #3: All comments have been addressed

2. Is the manuscript technically sound, and do the data support the conclusions?

Reviewer #3: Yes

3. Has the statistical analysis been performed appropriately and rigorously? 

Reviewer #3: Yes

4. Have the authors made all data underlying the findings in their manuscript fully available?

Reviewer #3: Yes

5. Is the manuscript presented in an intelligible fashion and written in standard English?

Reviewer #3: (No Response)

6. Review Comments to the Author

Reviewer #3: The method for G6PD testing was not updated in the abstract of the revised manuscript. Please revise Lines 39 & 40 in the abstract as indicated in CAPITAL LETTERS below:

Urine dipstick was used TO assay for the urine protein

and urine glucose, whilst the METHAEMOGLOBIN REDUCTION test was used for the G6PD deficiency .....

7. PLOS authors have the option to publish the peer review history of their article (what does this mean?). If published, this will include your full peer review and any attached files.

Reviewer #3: No

---

## [Editor Report · Acceptance letter]

23 Dec 2022

PONE-D-21-22935R3 

Clinical and laboratory presentation of first-time antenatal care visits of pregnant women in Ghana, a hospital-based study 

Dear Dr. Akonde:

I'm pleased to inform you that your manuscript has been deemed suitable for publication in PLOS ONE. Congratulations! Your manuscript is now with our production department. 

Kind regards, 

on behalf of

Dr. Orvalho Augusto 

Academic Editor

PLOS ONE